**Data Availability Statement:** All relevant data are within the paper and its Supporting Information files.

# Glucose deprivation reduces proliferation and motility, and enhances the anti-proliferative effects of paclitaxel and doxorubicin in breast cell lines *in vitro*

**Maitham A. Khajah** 📧*, **Sarah Khushaish, Yunus A. Luqmani**

Faculty of Pharmacy, Kuwait University, Kuwait, Kuwait

* maitham.khajah@ku.edu.kw, maitham@hsc.edu.kw

## Abstract

### Background

Breast cancer chemotherapy with high dose alkylating agents is severely limited by their collateral toxicity to crucial normal tissues such as immune and gut cells. Taking advantage of the selective dependence of cancer cells on high glucose and combining glucose deprivation with these agents could produce therapeutic synergy.

### Methods

In this study we examined the effect of glucose as well as its deprivation, and antagonism using the non-metabolized analogue 2-deoxy glucose, on the proliferation of several breast cancer cell lines MCF7, MDA-MB-231, YS1.2 and pII and one normal breast cell line, using the MTT assay. Motility was quantitatively assessed using the wound healing assay. Lactate, as the end product of anaerobic glucose metabolism, secreted into culture medium was measured by a biochemical assay. The effect of paclitaxel and doxorubicin on cell proliferation was tested in the absence and presence of low concentrations of glucose using MTT assay.

### Results

In all cell lines, glucose supplementation enhanced while glucose deprivation reduced both their proliferation and motility. Lactate added to the medium could substitute for glucose. The inhibitory effects of paclitaxel and doxorubicin were significantly enhanced when glucose concentration was decreased in the culture medium, requiring 1000-fold lesser concentration to achieve a similar degree of inhibition to that seen in glucose-containing medium.

### Conclusion

Our data show that a synergy was obtained by combining paclitaxel and doxorubicin with glucose reduction to inhibit cancer cell growth, which *in vivo*, might be achieved by applying

**Funding:** This work was supported by Kuwait University Research Sector grant PT02/18. Parts of this work were supported by grant SRUL02/13 to the Research Unit for Genomics, Proteomics and Cellomics Studies (OMICS), Kuwait University. the funders had no role in study design, data collection and analysis, decision to publish, or preparation of the manuscript.

**Competing interests:** The authors have declared that no competing interests exist.

a carbohydrate-restricted diet during the limited phase of application of chemotherapy; this could permit a dose reduction of the cytotoxic agents, resulting in greater tolerance and lesser side effects.

## Introduction

Nutritional imbalance, decreased physical activity, infection, stress, advanced age, and the use of chemotherapeutic agents and glucocorticoids may all contribute to the development of an hyperglycemic state in cancer patients [1, 2]. Several studies have reported correlation between various metabolic disorders characterized by hyperglycemia such as diabetes, and increased risk of breast cancer development and mortality [3–6] particularly in post-menopausal women [5]. This link is attributed in part to the increased utilization of glucose by cancer cells, and increase in the circulating levels of insulin-like growth factor-1 (IGF-1), which is also associated with increased risk of cancers [7–9].

Under prevailing aerobic conditions, normal cells derive their energy (ATP) primarily from oxidative phosphorylation of the products of glucose metabolism. Cancer cells however, even in the presence of adequate oxygen supply, seem to rely on the meagre output of ATP from glycolysis alone, with accumulation of lactate that is normally associated with the anaerobic state [10–13]. The upshot of this is that tumours are highly glucose-dependent and characterised by a high rate of glucose uptake and utilisation [14]. This phenomenon has long been known as the Warburg effect; it is the basis for the detection of metastatic tumour deposits by positron emission tomography with 2-deoxy glucose (2-DG). Many explanations have been proposed for this unusual preference, without a convincing resolution [15, 16]. This has not prevented attempts to use this metabolic anaomly to try and target cancer cells through glucose deprivation. Several reports have shown that administration of a low carbohydrate ketogenic diet to mice resulted in significant reduction of blood glucose and insulin levels, reduced breast cancer growth and metastasis and prolonged survival [17, 18]. In human cancer patients the outcome of similar studies has been ambivalent [19, 20], although such diets have been reported to successfully treat refractory drug-resistant epilepsy, obesity and type 2 diabetes [21–23].

Exposure of several breast cancer cell lines (MCF7, SUM-131502, T47D and ZR75-1), as well as cell xenografts grown in nude mice, to sodium-glucose transporter 2 inhibitors (canagliflozin and dapagliflozin) has shown inhibition of cell proliferation, in part through enhanced Amp-activated protein kinase (AMPK) phosphorylation and reduced phosphorylation of p70 ribosomal protein s6 kinase 1 (p70SK1) [24]. Prolonged treatment of mammary epithelial cells with transforming growth factor-beta (TGF-beta) was associated with enhanced expression of the glucose transporter Glut1 and glucose uptake, and subsequently enhanced cell proliferation [25]. This also induced a stable epithelial to mesenchymal transition (EMT), which is known to increase tumour cell aggressiveness [26, 27]. The expression of glucose transporter SLC2A1 and SLC2A3 was upregulated in MCF7 breast cancer cells that had undergone EMT as a result of shRNA induced estrogen receptor (ER) silencing [27]. EMT induced in MCF7 cells by exposure to increasing concentrations of tamoxifen also resulted in increased glucose consumption by the transited cells [28]. Another report suggested that SNAIL-1 (a downstream mediator of EMT) enhanced the gene expression patterns that promote glucose uptake and glycolysis [29]. The balance of these and other observations suggests that glucose deprivation may be an effective metabolic means of reducing tumour growth.

Current treatment of metastatic breast cancer, in particular of patients exhibiting *de novo* or *acquired* endocrine resistance, routinely involves the application of high dose chemotherapy with highly toxic agents whose adverse effects often outweigh their beneficial ones. The side effects of chemotherapy are dose-dependent which might be alleviated by reducing the doses, but this will ultimelty comprimise their efficacy.

In this study our aim was to determine whether a dose reduction in chemotherapeutic agents might be achieved by combining them with glucose deprivation, to create conditions most unfavourable to cancer cells.

## Materials and methods

### Cell lines

MDA-MB-231 human breast carcinoma cell line (ER–ve) was originally obtained from the ATCC (American Type Culture collection, VA, USA; catalogue number CRM-HTB-26). MCF10A normal breast epithelial cells were obtained from Dr. E Saunderson through Dr J Gomm St Bartholomews Hospital, London. pII (ER–ve) cells were generated by shRNA-mediated knockdown of the ER in MCF7 cells (which were also originally obtained from the ATCC, Catalogue number HTB-22). YS1.2 were also derived from similarly transfected MCF-7 cells [27, 30] but failed to down-regulate ER and therefore have been used as ER+ve.

Dulbecco's modified eagle's medium (DMEM) containing 25 mM glucose (ThermoFisher, USA, Cat# 12491) was used for regular culture. For experiments testing effects of glucose deprivation RPMI medium without glucose (Sigma-Aldrich, USA, Cat# R1383) was used. Both were supplemented with 5% fetal bovine serum (FBS), 600 mg/mL L-glutamine, 100 U/mL penicillin, 100 mg/mL streptomycin and 6 mL/500 mL 100 x non-essential amino acids. Glucose was obtained from Sigma Life Science, Cat # 49163). MCF10A were cultured in DMEM F12 (Cytiva, Cat# SH30023.01) supplemented with 5% horse serum, 1x Pen/Strep, 20 ng/mL mouse EGF, 0.5 μg/mL hydrocortisone, 100 ng/mL cholera toxin and 10 μg/mL insulin.

All cell lines were routinely grown in monolayer in 25 or 75 cm$^2$ tissue culture flasks (or in microtitre plates for experiments) inside an incubator maintained at 37˚C with 5% $CO_2$ atmosphere at 95% humidity. Cell cultures were periodically treated with mycoplasma removal agent from Biorad (USA) and tested with detection kits from Invivogen (CA, USA) and DAPI nuclear staining to ensure they remained free of mycoplasma.

### MTT assay

Cells were routinely seeded at $4x10^4$ into 24-well culture plates and allowed to grow to 30–35% confluency. The medium was then removed and replaced with fresh medium and additives according to individual experiments. Cell density was determined either immediately (day zero) or after 1 day and 4 days of cultivation. For the measurement, medium was removed and replaced with 500 μl of MTT reagent (Sigma-Aldrich, USA) (0.5 mg/ml) and left at 37˚C for 2 h; MTT solution was removed and 200 μl of acidic isopropanol added to dissolve the blue formazan crystals that had formed. Plates were scanned at 595 and 650 nm (for background subtraction) using a MULTISKAN SPECTRUM spectrophotometer, and absorbance compared between samples as a measure of proliferation. For measurement of growth over 30 days, cells were initially cultured in 24-well culture plates and transferred into larger vessels before confluency was reached. Cell growth in this case was assessed by trypsinising cells, centrifuging them and resuspending in a suitable volume of PBS, with aliquots taken for counting using a haemocytometer. Note that data have been presented directly as comparative OD readings, as the absolute cell number is not relevant for the purpose of this study. The actual cell density is indicated as the starting seeding number.

## Apoptosis assay

Cells were cultured in 6 well plates and were then trypsinized, pelleted by centrifugation at 1000g for 3 min and washed twice by re-suspension and centrifugation in ice-cold PBS and once in Annexin-V binding buffer (10 mM HEPES/NaOH (pH 7.4), 0.14 M NaCl, 2.5 mM CaCl$_2$). The final cell pellet was re-suspended in 100µl of Annexin-V binding buffer at $1\times10^6$ cells/ml and processed for FACS analysis using the PE Annexin V apoptosis detection kit I from BD Pharmingen (USA). Cells were stained in the following manner: (A), cells only (negative control) (B), with 10 µl of Annexin V-PE (C), with 20µl of 7AAD (D), with 10µl of Annexin V-PE plus 20µl 7AAD. All incubations were performed in the dark at room temperature (RT) for 15 min.

## Motility assay

Cells were cultured in 6 well plates with complete DMEM to 80–90% confluence. The medium was then aspirated off and replaced with a), DMEM containing 25 mM glucose (+ glucose) b), DMEM plus various concentrations of 2-DG (0.5–10 mM) c), RPMI without D-glucose (- glucose) d), RPMI without D-glucose plus various concentrations of added glucose (1.67–16.7 mM) or e), RPMI without D-glucose plus L-lactate (20 Mm, Sigma-Aldrich, USA, Cat # 71720). These media all contained FBS. A scratch was then created in the cell monolayer using a sterile p1000 pipette tip and a photograph of the scratched area was taken immediately (0 h). After incubation for 24h, further photographs were taken of the same scratched area. The width of the scratch at 24 h was calculated as a percentage of the width at 0 h; a minimum of 3 three areas along the scratch were measured and averaged, and experiments repeated three times.

## Lactate assay

Cells were cultured to a density of approximately $10^6$ in 6-well microtiter plates. The culture medium was carefully aspirated into Eppendorf tubes and protein concentration was estimated using the Bradford assay. Extracellular lactate was measured in aliquots using the Enzy-Chrom L-Lactate Assay Kit ECLC-100 purchased from BioAssay Systems USA, following the manufacturer's protocol. Standards were prepared by dilution of a stock solution of 100 mM L-lactate in serum free media, and 20 µl of samples or standards were transferred into wells of a clear bottom 96-well plate. Two reactions were performed for each sample: one with both enzymes A and B, and another without enzyme A (control). The working reagent was prepared freshly by mixing 60 µl Assay Buffer, 1 µl enzyme A, 1 µl enzyme B, 10 µl NAD and 14 µl MTT. For control, enzyme A was omitted from the reagent mix; 80 µl of the working reagent was added to each sample well and mixed by pipetting up and down. The background optical density at 650 was measured in a plate reader at 'zero' time ($OD_0$) and after 20 min ($OD_{20}$) incubation at room temperature and subtracted from that at 565nm. For standard curve the corrected $OD_0$ was subtracted from $OD_{20}$. For samples with no enzyme A control, the $\Delta OD_{\text{no enzA}}$ value was subtracted from $\Delta OD_{\text{sample}}$. The $\Delta\Delta OD$ values were used to determine sample L-lactate concentration from the standard curve.

## Statistical analysis

Student's two tailed unpaired t- test or one-way ANOVA test followed by Bonferroni post hoc test were used to compare means of individual groups using GraphPad Prism software (version 5.0): $p < 0.05$ was considered statistically significant.

## Results

### Effect of glucose starvation on cell proliferation

We wanted firstly to determine whether glucose starvation reduces normal and breast cancer cell proliferation. As shown in Fig 1A and 1B, culturing of both ER–ve pII cells as well as ER +ve YS1.2 cells in culture medium without glucose inhibited their proliferation (but did not induce cell apoptosis; S1 Fig) after 4 days (but not 1 day) of culture. This was also seen with the other ER–ve MDA-MB-231 cells (Fig 1C) and the normal breast epithelial cell line MCF10A (Fig 1D). In another set of experiments, performed on the MCF10A and pII cells, we determined the longer-term effect (up to 30 days) of alternating culture medium with or without glucose every 72h. For MCF10A, a significant increase in growth rate was observed over the entire period of 30 days in the presence of glucose, while glucose starvation or intermittent supply of glucose both completely suppressed growth over the whole period but maintained them at around seeding level (Fig 1E). For pII cells, a significant increase in growth rate was also observed over the 30 days when glucose was supplied while complete glucose starvation killed the cells from day 8 onwards (Fig 1F). Periodic supply of glucose to pII cells every 72h allowed limited growth, though to a much lesser extent than with a continuous supply of glucose. These data suggest that glucose starvation from the culture medium inhibited cell proliferation.

### Effect of glucose concentration on breast cancer cell proliferation and motility

Since glucose starvation significantly inhibited cell proliferation, we determined whether growth could be stimulated by supplying increasing amounts of glucose. As shown in Fig 2, glucose supplementation to cancerous (A-C) and normal (D) breast cell lines increased their proliferative rate in a concentration-dependent manner. Supplying exogenous glucose to concentrations (17 mM) similar to those in DMEM containing glucose (+ glucose) restored cell proliferation to a similar degree. This effect was also seen with cell motility; glucose starvation significantly reduced ER–ve (MDA-MB-231 and pII, Fig 3A–3C) and ER +ve (YS 1.2, Fig 3D) breast cancer cell motility whereas it was enhanced by glucose supplementation in a concentration-dependent manner. These data suggest that glucose supplementation enhances cell proliferation and motility in a concentration-dependent manner.

### Effect of glucose starvation on production of lactate

Herein, we wanted to determine whether extracellular lactate level is modulated in the ER -ve breast cancer cells pII when cultured in medium without glucose. Lactate was measured in the culture medium from the incubation of pII cells grown in medium with or without glucose for 24h. The data in Fig 4 shows that cells cultured in medium without glucose secreted almost no lactate, while cells cultured in medium containing glucose secreted lactate that reached concentrations of approximately 20 mM under the conditions examined.

### Effect of adding glucose or lactate to glucose starved cells on their motility

Next, we wanted to determine the effect of supplying exogenous glucose to cells cultured in medium without glucose, and whether lactate can substitute for glucose in the culture medium to enhance cell motility. The data in Fig 5 show that the motility of pII cells (assessed using the wound closure assay) after 24h in medium without glucose was much reduced (panel B;—glucose,) compared to cells grown in medium with glucose (panel A; + glucose,); however, when incubation was continued for a further 24h after changing the—glucose medium to medium

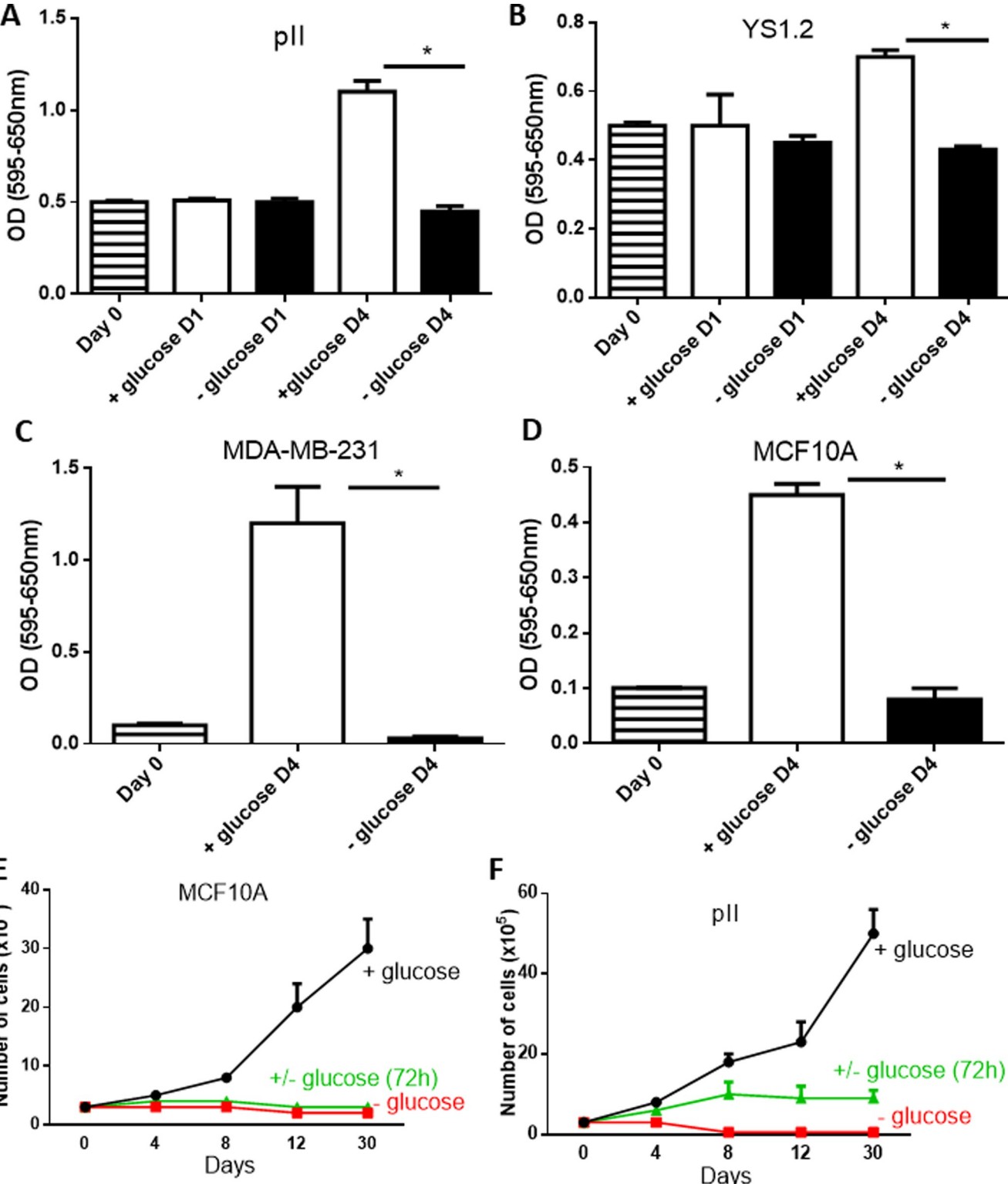

**Fig 1. Effect of glucose starvation on cell proliferation.** pII (panel A) and YS1.2 (panel B) cell density was determined, using the MTT assay, at seeding day (day 0, hatched bars), and at days 1 and 4 (D 1 and D4) after culture in medium containing glucose (open bars) or without glucose (solid bars). The degree of proliferation of MDA-MB-231 and MCF10A as indicated in panels C-D was determined at day 4 (D 4) after culture in medium with (open bars) or without (closed bars) glucose. Panel E (MCF10A) and panel F (pII), show growth of cells (number of cells were measured using hemocytometer) cultured in + glucose medium (black line),—glucose medium (red line), or these two media alternated every 72 h (green line). Histobars represent means ± SEM of at least 3 independent determinations. * denotes significant difference from cells cultured in + glucose medium with p<0.05.

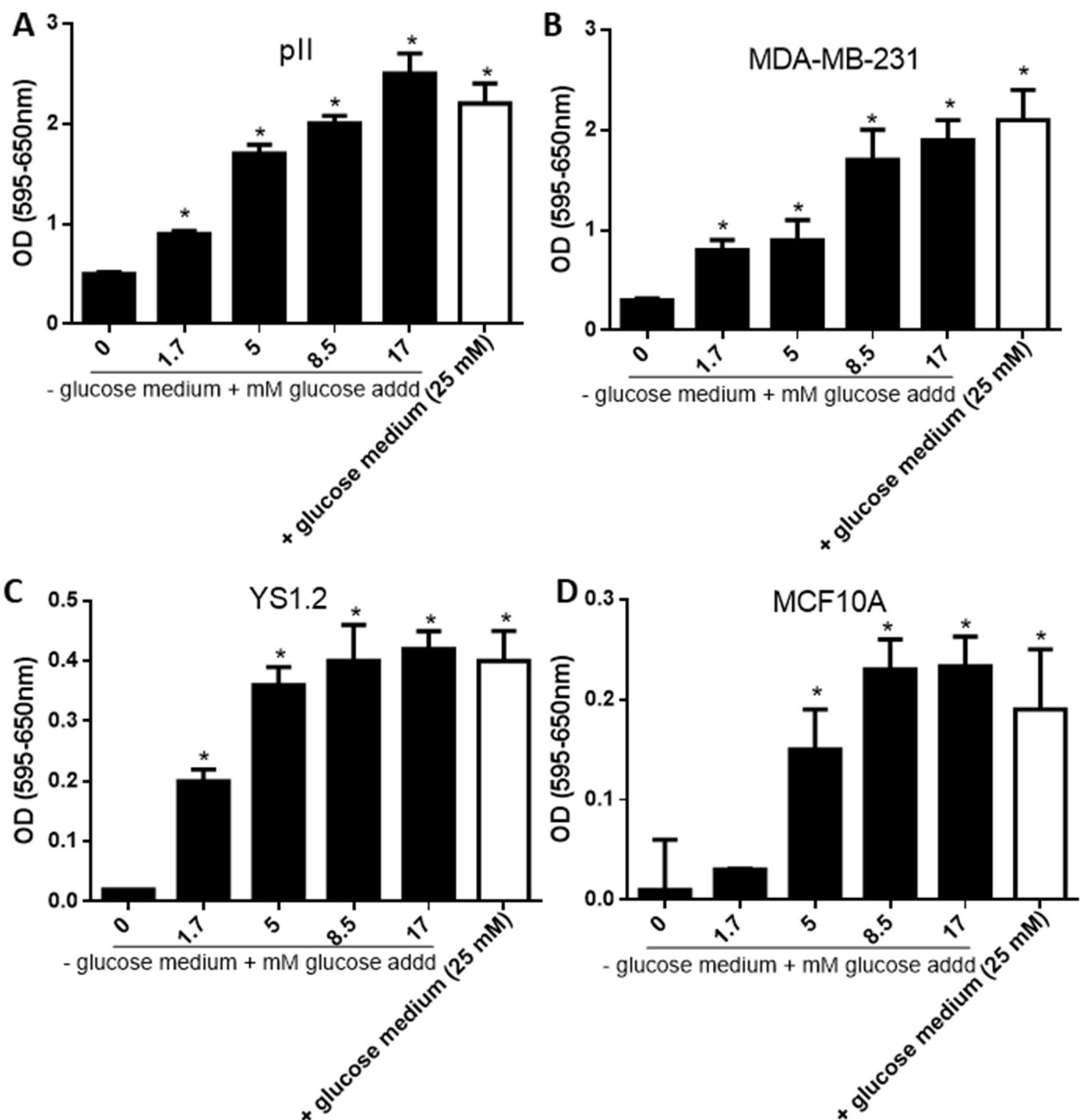

**Fig 2. Effect of glucose concentration on breast cancer cell proliferation.** Proliferation of pII (panel A), MDA-MB-231 (panel B), YS1.2 (panel C), and MCF10A (panel D) cells after culture for 4 days in + glucose medium (open bars) or–glucose medium supplemented with various concentrations of glucose as indicated (solid bars), was determined using the MTT assay. Histobars represent means ± SEM of at least 3 independent determinations. * denotes significant difference from cells cultured in—glucose medium, with $p < 0.05$.

containing glucose or to glucose-free medium containing 20 mM lactate, the cells regained their motility (panel B). Combined data from 3 independent experiments is given in panel C.

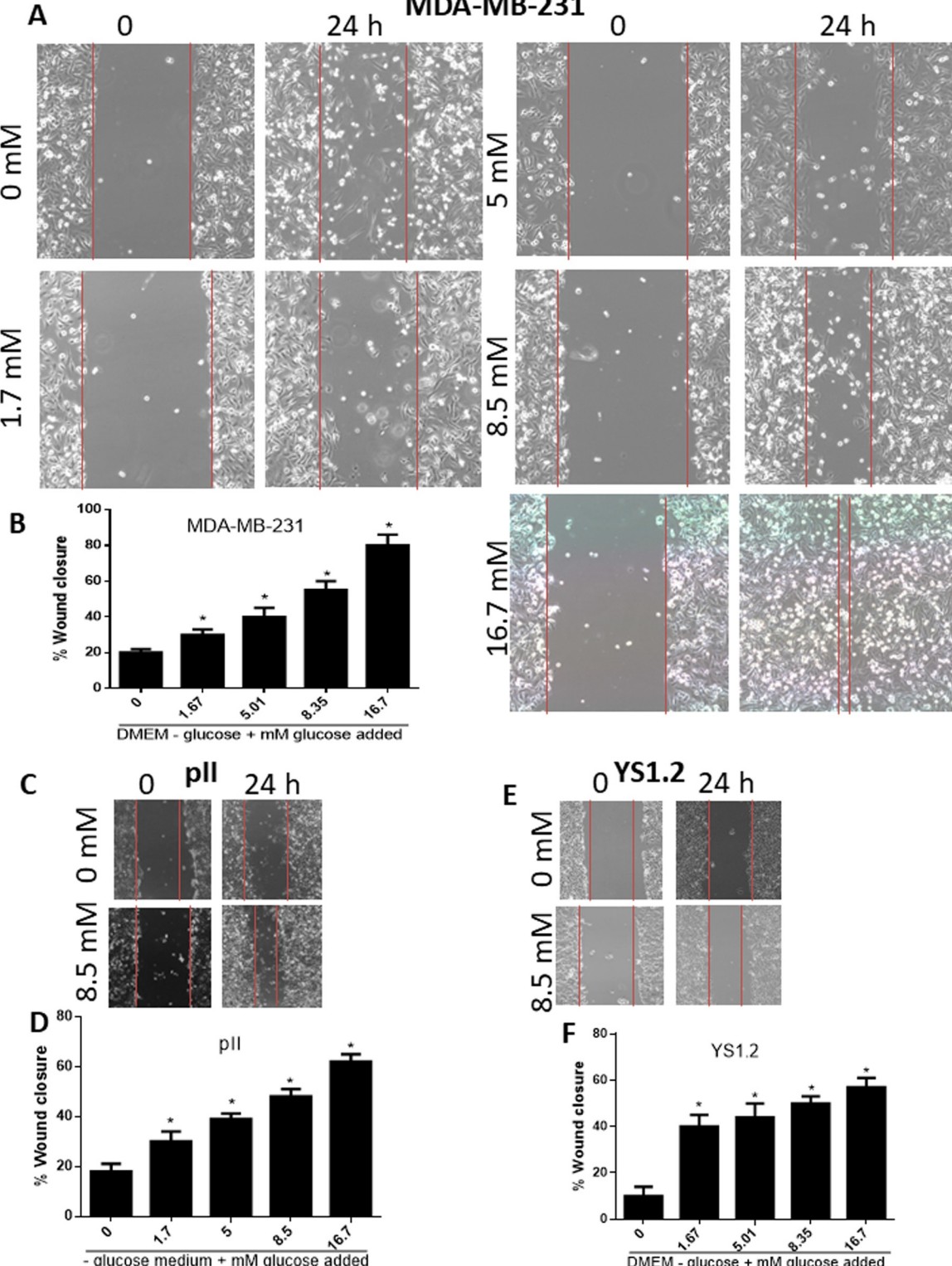

**Fig 3. Effect of glucose concentration on breast cancer cell motility.** Cells were cultured to confluency in + glucose medium. The medium was then changed to–glucose medium + glucose additions as indicated. A scratch was made through the cell monolayer and the width measured immediately and after further 24h incubation. Panels A-B for MDA-MB-231 cells, panels C-D for pII cells, and panels E-F for YS1.2 cells. Histobars represent means ± SEM for each condition. * denotes significant difference from cells cultured in—glucose medium, with $p<0.05$.

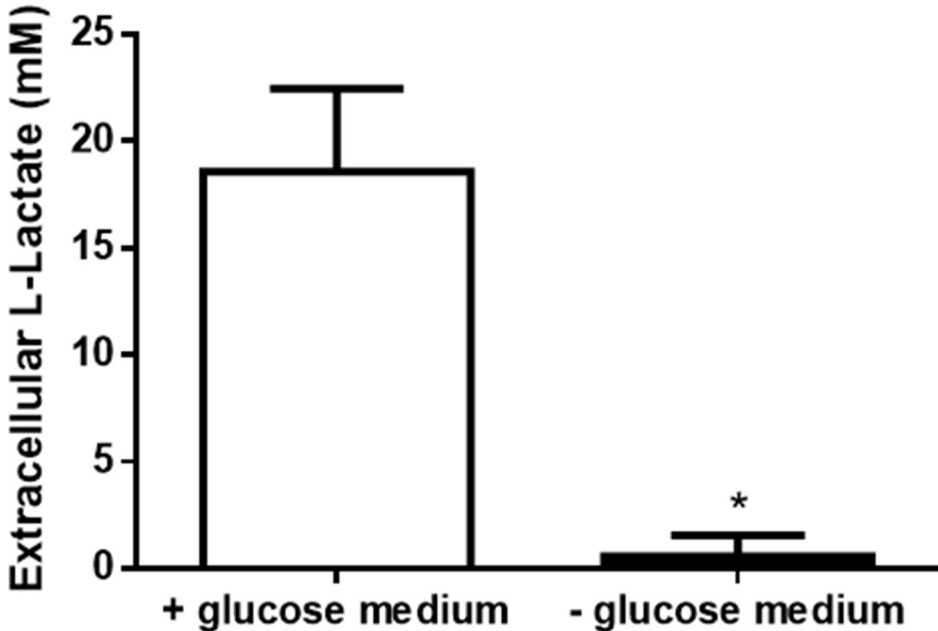

**Fig 4. Effect of glucose starvation on lactate levels.** Extracellular lactate level in pII cells upon culture in + glucose medium (open bar), or—glucose medium (solid bar) for 1 day was determined as described in Methods. Histobars represent means ± SEM of at least 3 independent determinations. * denotes significant difference from cells cultured in + glucose medium, with $p<0.05$.

## Effect of 2-deoxy glucose on cell proliferation and cell motility

2-DG is a D-glucose analogue which inhibits glycolysis through formation and intracellular accumulation of 2-deoxy-D-glucose-6-phosphate, a competitive inhibitor of hexokinase and glucose-6-phosphate isomerase [31]. The breast cancer cell lines were plated in culture medium with glucose and then after 24h exposed to increasing concentrations of 2-DG for 4 days. MTT assay performed on the cells showed that 2-DG dose-dependently inhibited cell proliferation in all cell lines, with significant decrease starting at concentrations from 0.5–1 mM and reaching substantial inhibition by 10 mM, the highest concentration tested (Fig 6). 2-DG also dose-dependently inhibited both ER +ve and ER–ve breast cancer cell motility (performed using scratch assay, Fig 7).

## Glucose depletion significantly enhanced the anti-proliferative effects of paclitaxel and doxorubicin in breast cancer cell lines

We wanted to determine if glucose depletion enhances the sensitivity of paclitaxel or doxorubicin in inhibiting cell proliferation. Paclitaxel (Fig 8) and doxorubicin (Fig 9) treatment significantly reduced cell proliferation in a concentration dependent manner. There was a significant inhibition in pII cell proliferation at 1 µM paclitaxel, which was further increased to 90% with 10–100 µM (Fig 8A). For MDA-MB-231 cells, paclitaxel inhibited cell proliferation (by 50%) with 0.1 µM, which reached 98–99% inhibition with 1–100 µM (Fig 8B). For YS 1.2 cells, paclitaxel (0.1–100 µM) inhibited cell proliferation by 80% (Fig 8C). For MCF10A, 20–25% inhibition was seen with paclitaxel at concentrations of 0.1–1 µM and reached 50% inhibition at concentrations of 10–100 µM (Fig 8D). Of note, breast cancer cells were more sensitive to paclitaxel treatment when compared to the normal epithelial cell line MCF10A. Doxorubicin significantly inhibited pII cell proliferation (by 30%) at a concentration of 100 µM (Fig

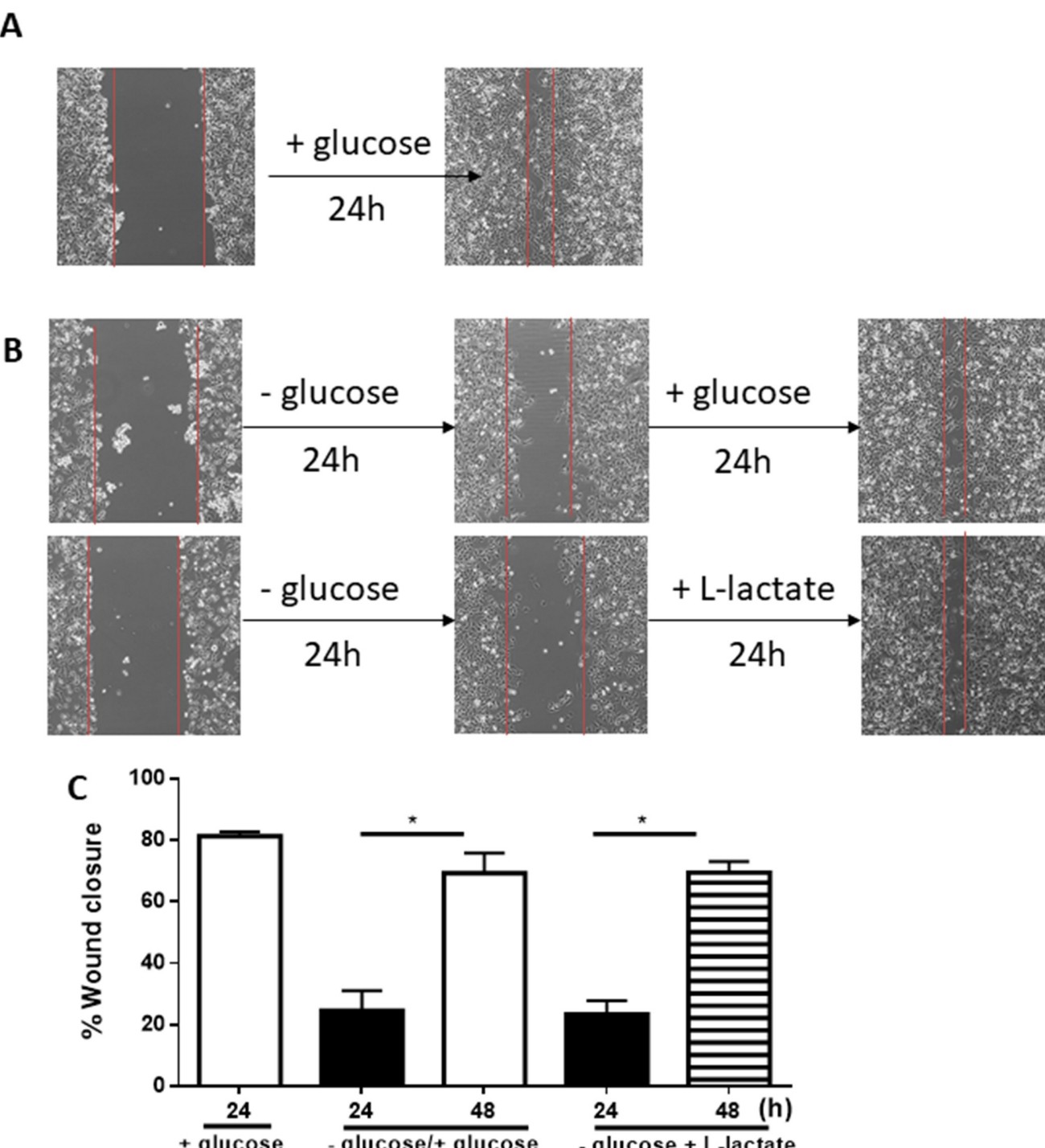

**Fig 5. Effect of glucose starvation on pII cell motility.** The degree of pII cell motility upon culture in + glucose medium (panel A),—glucose medium (panel B), -glucose medium for 24 h followed by culture in +glucose medium for another 24 h, or -glucose medium in the presence of 20 mM L-lactate was determined (panels B-C) as described in Methods. Histobars represent means ± SEM of at least 3 independent determinations. * denotes significant difference with p<0.05.

9A). For MDA-MB-231 cells, doxorubicin significantly inhibited cell proliferation (by 80–99%) at all the concentrations used (Fig 9B). These data suggest that *de novo* resistant breast

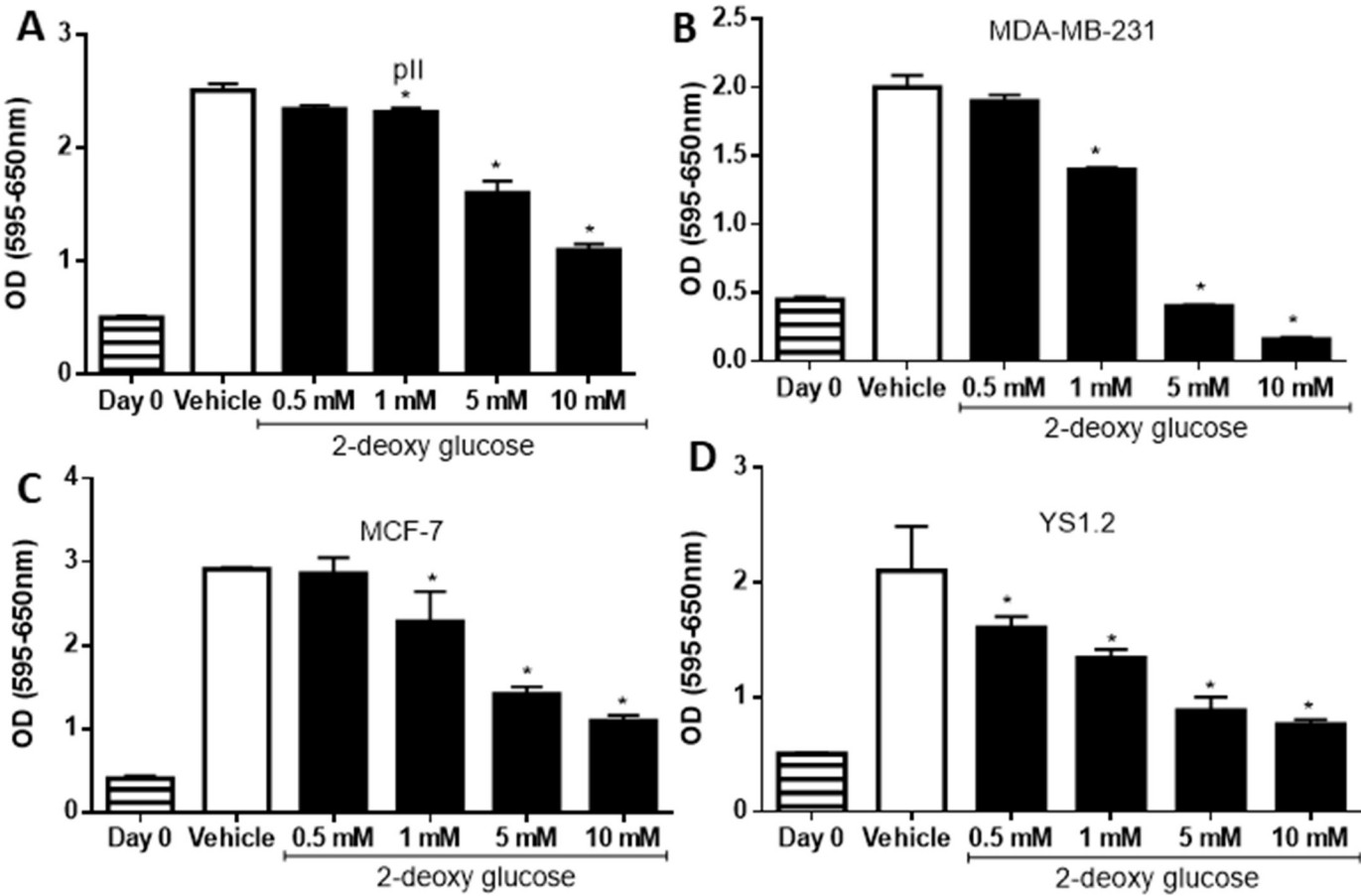

**Fig 6. Effect of 2-deoxy glucose on cell proliferation.** Proliferation of the cell lines indicated was measured after 4 days of exposure to either vehicle or various concentrations of 2-DG, using the MTT assay. Histobars represent means ± SEM of at least 3 independent experiments. * denotes significant difference from vehicle-treated cells, with $p < 0.05$.

cancer cells are more sensitive to the anti-proliferative effect of doxorubicin when compared to the *acquired* endocrine resistant cells. For YS 1.2, there was a significant inhibition in cell proliferation (by 50%) at 1 μM, which further increased to 70–80% with 10–100 μM (Fig 9C). For MCF10A, there was a significant inhibition of cell proliferation (by 40–50%) at the concentrations used (Fig 9D).

To determine whether a synergistic/additive effect could be obtained in low glucose medium, we added these drugs to cells cultured in either medium with/without glucose [vehicle treated cells cultured in glucose containing medium (25mM) was taken as 100%] or supplemented with low (1.7 or 5 mM) glucose. When the tested cell lines were cultured in medium without glucose (for 4 days), there was no evidence of cell proliferation; adding either paclitaxel or doxorubicin was of no benefit (data not shown). Therefore, we next determined the effect of these two agents in a medium supplemented with low glucose.

For pII cells, glucose reduction significantly enhanced the anti-proliferative effects of both paclitaxel and doxorubicin when compared to treatment in the presence of glucose. The addition of 1.7 mM glucose had a greater enhancement effect than the higher amount. In the reduced glucose medium, treatment with 0.1 μM paclitaxel induced a broadly comparable degree of inhibition to that achieved with 100 μM in the presence of glucose in the culture medium (Fig 8A). Also, in reduced glucose medium, treatment with 0.1 μM of doxorubicin

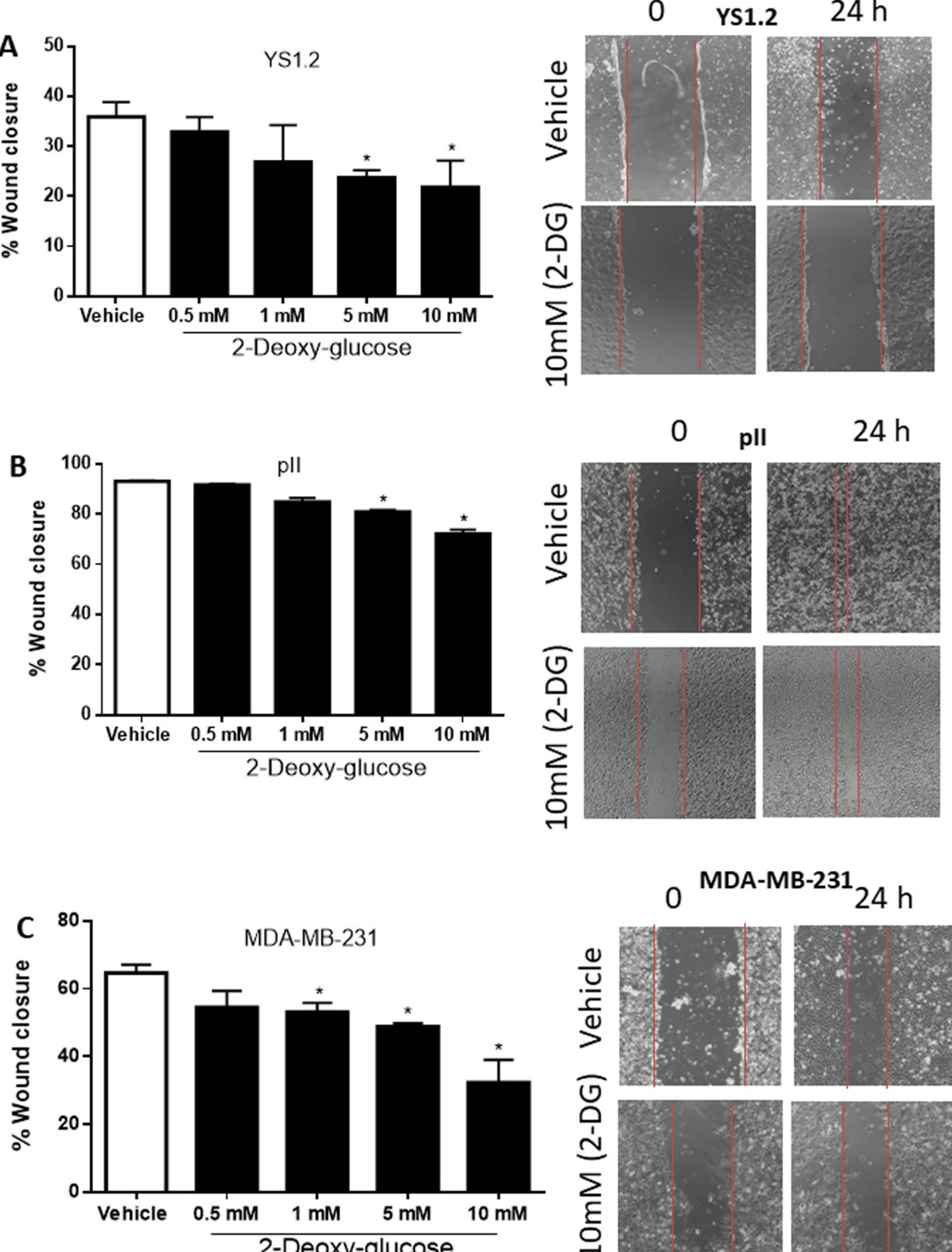

**Fig 7. Effect of 2-deoxy glucose on cell motility.** Motility of the YS1.2 (A), pII (B), and MDA-MB-231 (C) cells in response to treatment with various concentrations of 2-DG or vehicle, was measured using the wound healing assay. Histobars represent means ± SEM of at least 3 independent experiments. * denotes significant difference from vehicle treated cells, with p<0.05.

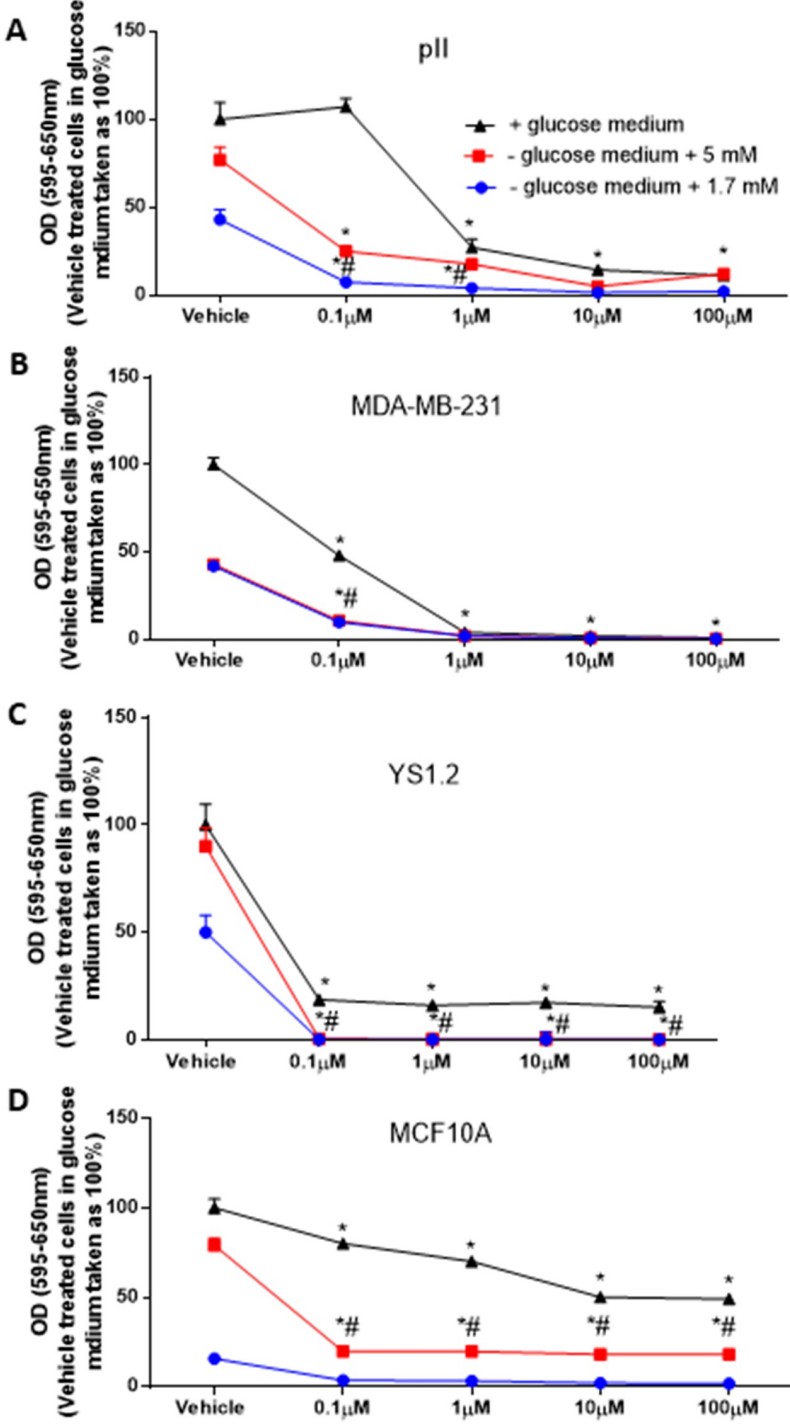

**Fig 8. Effect of glucose depletion on the anti-proliferative effects of paclitaxel in breast cancer cell lines.** The effect of paclitaxel on cell proliferation upon culture in +glucose medium (black line, taken as 100%) or -glucose medium plus 5 mM (red line), or 1.7 mM glucose (blue line) was determined at day 4 using the MTT assay. Histobars represent means ± SEM of at least 3 independent determinations. * denotes significant difference from cells treated with vehicle (normal saline), with $p < 0.05$.

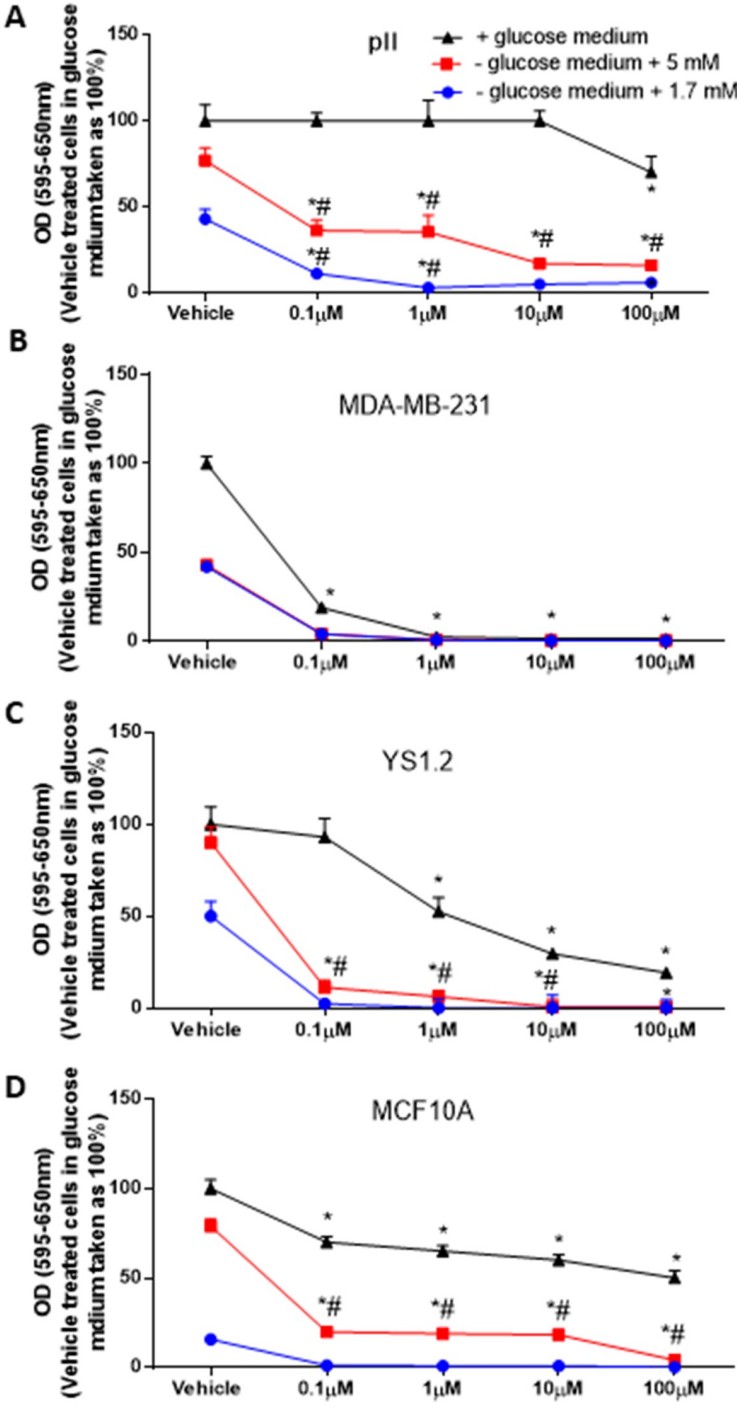

**Fig 9. Effect of glucose depletion on the anti-proliferative effects of doxorubicin in breast cancer cell lines.** The effect of doxorubicin on cell proliferation upon culture in +glucose medium (black line, taken as 100%) or—glucose medium plus 5 mM (red line), or 1.7 mM glucose (blue line) was determined at day 4 using the MTT assay. Histobars represent means ± SEM of at least 3 independent determinations. * denotes significant difference from cells treated with vehicle (normal saline), with $p < 0.05$.

induced a greater degree of inhibition than 100 μM concentration in the presence of glucose in the culture medium (Fig 9A).

For MDA-MB-231 cells, glucose reduction enhanced the anti-proliferative effect of paclitaxel (but not doxorubicin) at 0.1 μM (Figs 8B and 9B) and higher concentrations of both agents abolished cell proliferation in all of the tested conditions. For YS1.2, higher degree of inhibition at all paclitaxel concentrations was achieved in cells cultured in glucose reduced medium (Fig 8C), and a similar degree of inhibition with doxorubicin was achieved with 0.1 μM in glucose reduced medium compared with 100 μM in glucose containing medium (Fig 9C). Similar effects were also seen with MCF10A cells (Figs 8D and 9D).

## Discussion

In the current report, we present experimental evidence for involvement of glucose in cell proliferation and motility in several breast cell lines. The effect of glucose deprivation on cell proliferation was more profound in ER–ve when compared to ER +ve breast cancer cells. This might be explained by their higher proliferative rate which required higher glucose concentrations (Figs 1 and 6). Both of these processes were significantly enhanced by glucose in a concentration dependent manner in both cancer and normal cells. Conversely, they were inhibited by glucose starvation or competing out the glucose in culture medium by addition of the non-metabolizable analogue 2-DG. Furthermore, we observed a synergistic inhibitory effect on proliferation by combining glucose deprivation (even if applied periodically) with addition of two commonly used agents for breast cancer chemotherapy, paclitaxel and doxorubicin. In low glucose conditions, the concentration of both of these drugs could be reduced several-fold to achieve the same degree of inhibition seen in the high glucose medium. If this can be reproduced *in vivo*, it would greatly facilitate the utilization of these highly toxic agents by allowing much lower doses to be applied to achieve the same therapeutic effect. This would mean significant reduction in side effects and greater tolerance. Preliminary experiments to explore the mechanism whereby glucose stimulates growth and motility suggest that it may, at least in part, be mediated through lactate secreted into the extracellular environment.

Hyperglycemia has been shown to contribute to enhanced breast cancer cell proliferation, metastasis and chemotherapy resistance [8, 32–34]. Furthermore, glucose and other factors involved in glucose metabolism such as insulin and insulin like growth factors enhance breast cancer cell proliferation and contribute to breast cancer development [35–38]. High glucose concentration (25–50 mM) significantly enhanced MCF-7 and T47D cell proliferation and decreased cell apoptosis and necrosis, while low glucose concentration (2.5 mM) induced cell apoptosis and necrosis [37]. In addition, high glucose concentration (25 mM) in the culture medium of MCF-7 and T47D enhanced IGF-1 induced cell growth; this was not observed when cells were cultured in low glucose containing medium (5 mM) [39]. These data are in agreement with our own observations, where the addition of increasing concentration of glucose to the culture medium significantly enhanced breast cancer cell proliferation (Fig 2) and motility (Fig 3). Adding 2-DG was effective in blocking both proliferation and motility. Several previous reports have demonstrated similar effects of 2-DG on breast cancer cells, with reduction in their migration [40] and proliferation [41–45]. In another recent report, enhanced glucose uptake was correlated with enhanced invasiveness of several breast cancer cell lines, whereas low doses of 2-DG (1 mM) significantly reduced their invasive capacity [46]. Our data is consistent with these reports, in the same concentration range (0.5–1 mM). Chen *et al* [47] reported that short term glucose deprivation (24 h) in various breast cancer cell lines induced cell death, which was higher in MDA-MB-231 than MCF-7 cells, in part through enhanced AMPK phosphorylation. This was also observed in another report using MCF-7 and T47D cells [48]. We did not see much difference in cell proliferation at 24 h; it was evident at day 4 of glucose starvation in both the cancer lines as well as the normal cells. Periodic re-supply of

glucose (every 72 h; Fig 1E and 1F) maintained the cells at the seeding number for MCF10A and facilitated partial growth for pII cells.

The unresolved issue of why cancer cells prefer anaerobic metabolism (and hence the high glucose requirement) has given rise to speculation that this might somehow confer advantages in growth through increased production of precursors of nucleic acid biosynthesis through the pentose phosphate shunt and faster (if less efficient) production of ATP [49]. The accumulation of lactate as the end product of glucose catabolism could also be an important factor but has so far not received quite as much attention. To prevent cellular acidosis, excessive lactate is thought to be secreted into the extracellular environment from where it has been proposed that it may be taken up and used as a metabolic substrate by other aerobically active cells by re-conversion to pyruvate [50]. The concurrent acidification of the extracellular space by co-transport of H+ with the lactate extrusion is where other researchers have focused, leading to suggestions that these acidic conditions promote migration/metastasis [51, 52]. An alternative possibility is that the increased aggressiveness is actually not due to the lowered pH but instead to the elevated extracellular lactate that is formed as a result of excessive glycolytic activity. Thus, the effect of glucose in promoting cancer cell proliferation/motility, which we and many others have observed, could be mediated by lactate. Providing lactate to pII cells deprived of glucose (without any change in pH) had the same effect as giving back glucose. It is uncertain how exactly extracellular lactate exerts its effect although we do have some preliminary data suggesting modulation in the activity of signaling pathways; however, we do not think there is any direct connection between this and the action of Dox/Paclitaxel. It is simply that lactate inhibition allows use of much lower concentrations of CTX drugs (to 'finish off' the cancer cells), and that this could be simply achieved by a glucose restrictive diet. We are currently studying this phenomenon further. Somewhat surprisingly, there was no clear difference in dependence on glucose between the cancer cell lines and the normal MCF10A; we had expected the latter to have been less reliant on glucose and possibly have been able to utilise other substrates, principally the glutamine presents in the culture medium, for energy generation through oxidative phosphorylation. Perhaps this highlights a disadvantage of *in vitro* models. *In vivo*, normal cells are known to revert to utilisation of fatty acids under glucose limiting conditions [53].

As already discussed, cancer cells mainly utilize aerobic glycolysis as the main route of glucose metabolism after glucose entry into cells through glucose transporters. This results in the generation of various glucose metabolites which may enter other metabolic pathways such as the serine/glycine and pentose phosphate metabolic pathways (termed non-glycolysis metabolic pathways). All the products generated through these various pathways play an important role in cancer metabolism, progression, and metastasis [54, 55]. The pentose shunt provides precursors for nucleic acid biosynthesis that is important for cancer cells to proliferate. It was demonstrated that the expression profile of serine/glycine metabolic pathway is unique to the molecular subtype of breast cancer, with high activity in HER-2 [56] and triple negative types [57], which is correlated with poor clinical prognosis [58]. In addition, enhanced expression profile of the pentose phosphate pathway-related enzymes such is 6PGDH and TKT is also evident in breast cancer [59, 60], especially in HER-2 and triple negative types [61], which is again associated with poor clinical prognosis [62–64].

The other major aspect of this study was to determine whether glucose deprivation could be used to enhance the effectiveness of chemotherapy. Both paclitaxel and doxorubicin are commonly used to kill cancer cells but their wider toxicity at the necessary therapeutically high concentration also affects some normal tissues such as immune, gut and hair cells. Our data shows that if they are added to glucose starved cells, their effective concentration can be very significantly lowered, giving obvious advantages. As a therapeutic approach, depriving the

body completely of glucose would of course have serious physiological consequences but lowering it temporarily only whilst chemotherapy was being administered might be tolerated. One way to achieve this situation is through the use of ketogenic diet (KD) for a short period of time. The effect of KD on metabolic parameters is variable depending on the composition of protein, fat, and carbohydrates as well as on the duration of its use. Some reports suggested that KD was shown to reduce fasting blood glucose and HbA1c levels in patients with diabetes [65–68], in normal subjects [69] and in rodents [70]. Interestingly, KD was also shown to reduce glucose metabolism in rodents and humans [71–73]. There is extensive clinical evidence indicating that KD is well tolerated in patients with various forms of cancer, resulting in improvement in quality of life with no serious adverse events or toxicity reported [74–87]. KD was shown to enhance the anti-tumor effects of various agents as well as when combined with radiotherapy. Combination of KD with bevacizumab in an orthotopic U87MG glioblastoma model in nude mice increased survival rate when compared with bevacizumab monotherapy [19]. Two reports suggested that KD significantly enhanced the anti-tumor and anti-angiogenic effects of metronomic cyclophosphamide in neuroblastoma xenografts in a CD1-nu mouse model [88, 89]. Furthermore, another report demonstrated that KD combination with radiation or carboplatin led to slower tumor growth in mice bearing NCI-H292 and A549 lung cancer xenografts; in part through increased oxidative damage mediated by lipid peroxidation [90]. KD was also shown to improve responses to several PI3K inhibitors in tumors with a wide range of genetic aberrations such as in patient-derived xenograft models of advanced endometrial adenocarcinoma (harboring a *PTEN* deletion and *PIK3CA* mutation), bladder cancer (*FGFR*-amplified), in syngeneic allograft models of *PIK3CA* mutant breast cancer and in a *MLL-AF9* driven acute myeloid leukemia. Combination of KD improved the efficacy of several agents which target the PI3K pathway through inhibition of insulin feedback (which limits the efficacy of these agents) in part through decreased cell proliferation and increased apoptosis [17].

## Supporting information

**S1 Fig. Effect of glucose starvation on cell apoptosis.** pII (solid bars) and YS1.2 (open bars) were cultured for 1 and 4 days in medium containing glucose (+ glucose) or without glucose (-glucose). Cell apoptosis was determined by flow cytometry using Annexin-V/7AAD staining as described in the methods. Histobars represent means ± SEM of at least 3 independent determinations.
(TIF)

## Author Contributions

**Conceptualization:** Maitham A. Khajah, Yunus A. Luqmani.

**Data curation:** Maitham A. Khajah, Sarah Khushaish.

**Formal analysis:** Maitham A. Khajah, Sarah Khushaish.

**Funding acquisition:** Maitham A. Khajah.

**Investigation:** Maitham A. Khajah, Yunus A. Luqmani.

**Methodology:** Maitham A. Khajah, Sarah Khushaish.

**Project administration:** Maitham A. Khajah.

**Resources:** Maitham A. Khajah, Yunus A. Luqmani.

**Supervision:** Maitham A. Khajah.

**Writing – original draft:** Maitham A. Khajah.

**Writing – review & editing:** Maitham A. Khajah, Yunus A. Luqmani.

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
