## [Decision Letter · Decision Letter 0]

26 May 2022

PONE-D-22-03147Glucose deprivation reduces proliferation and motility, and enhances the anti-proliferative effects of paclitaxel and doxorubicin in breast cell line in vitroPLOS ONE

Dear Dr. Khajah,

Thank you for submitting your manuscript to PLOS ONE. After careful consideration, we feel that it has merit but does not fully meet PLOS ONE’s publication criteria as it currently stands. Therefore, we invite you to submit a revised version of the manuscript that addresses the points raised during the review process.

We look forward to receiving your revised manuscript.

Kind regards,

Yi-Hsien Hsieh, Ph.D.

Academic Editor

PLOS ONE

Journal Requirements:

This work was supported by Kuwait University Research Sector grant PT02/18. Parts of this work were supported by grant SRUL02/13 to the Research Unit for Genomics, Proteomics and Cellomics Studies (OMICS), Kuwait University. 

This work was supported by Kuwait University Research Sector grant PT02/18. Parts of this work were supported by grant SRUL02/13 to the Research Unit for Genomics, Proteomics and Cellomics Studies (OMICS), Kuwait University. 

the funders had no role in study design, data collection and analysis, decision to publish, or preparation of the manuscript.

Reviewers' comments:

Reviewer's Responses to Questions

**Comments to the Author**

1. Is the manuscript technically sound, and do the data support the conclusions?

Reviewer #1: Yes

Reviewer #2: Partly

Reviewer #3: Yes

2. Has the statistical analysis been performed appropriately and rigorously? 

Reviewer #1: Yes

Reviewer #2: Yes

Reviewer #3: Yes

3. Have the authors made all data underlying the findings in their manuscript fully available?

Reviewer #1: Yes

Reviewer #2: Yes

Reviewer #3: Yes

4. Is the manuscript presented in an intelligible fashion and written in standard English?

Reviewer #1: No

Reviewer #2: Yes

Reviewer #3: Yes

5. Review Comments to the Author

Reviewer #1: The authors examined the effect of glucose deprivation on growth and proliferation of breast cells, and the potentiation of Glucose deprivation on anti-proliferative effects of paclitaxel and doxorubicin. The novalty of this manuscript should be improved by exploring the underlying mechanism. As such, it still has some clinical value for drug treatment of tumors. A few of suggestions are shown below:

1. If possible, the author should provide some lines of supportive evidence from animal experiments. However, it is hard to set up a low-glucose condition, because tumor cells have a strong ability of glucose uptake to enter the glycolysis.

2. in Figure 8, different concentrations of 2-DG treatment caused the similar effects to those obtained from Glucose deprivation. Also, addition of lactate or pyruvate could offset the effect of glucose deprivation.

3. The authors should provide some mechanistic evidence to support their conclusion.

Reviewer #2: The manuscript: Glucose deprivation reduces proliferation and motility, and enhances the anti-proliferative effects of paclitaxel and doxorubicin in breast cell line in vitro, brings information already available in the literature on the effects of glucose metabolism on tumor and normal cell proliferation. Tumor cells produce a high level of reactive oxygen species compared to normal cells due to the increased activation of various metabolic pathways. Glycolysis by the Warburg effect maintains redox homeostasis by being independent of mitochondrial oxidative phosphorylation that produces a large amount of ROS, increasing metabolic secondary pathways, NADPH, G6PD and 6-Phosphogluconate dehydrogenase, which produces an effect on tumor metabolic reprogramming.

The data presented on the cytotoxic activity of the conditions of glucose deprivation, addition of lactate, DG2gly, should be normalized, since the data are presented in optical density values and do not represent the biological effects after the different experimental conditions. Thus information about the effects of cytotoxicity can be better interpreted and included in the discussion. The data presented in the supplementary material does not demonstrate the antiproliferative effects of the metabolic effects of glucose deprivation, since no increase in cells in apoptosis or necrosis was found.

Include include in the discussion other aspects of non-glycolytic metabolism (Non-glycolysis Glucose Metabolism Pathway in Breast Cancer). The authors could apply a predictive pharmacological test to demonstrate the synergistic effects of chemotherapeutic combinations in deprivation or supplementation conditions, such as additive, anatoagonic or synergistic effects.

Reviewer #3: Minor Issues:

Materials and methods- Cell lines- second paragraph:

* The authors wrote 100 U/Ml penicillin should be corrected to 100 U/mL

* RPMI medium used (Cat# R1383) contains L-glutamine (0.3 g/L) and you also added L-glutamine as a supplement.

This could be another source of energy for the cells. L-glutamine serves as an auxiliary energy source, especially when

cells are rapidly dividing.

* Why the non-essential amino acids were added to the culture medium? Published research reported that non-essential

amino acids attenuate apoptosis.

* According to the manufacturer’s instructions the concentration of cells for Annexin V apoptosis experiment is 1 x 10*6

cells/ml, why the authors used a concentration of 5 x 10*6 cells/ml.

* Why the authors did not use the same medium for the experiments with glucose and those without glucose?

Figure 1: To denote a significant difference, please insert a line above the bars that are targeted and add a * above the line.

This will make it easier for the readers to follow.

Figure 1: I think the X axis in the figure can be labelled in a more appropriate way.

Figure 3: It is not mentioned in the legend and neither on the figure the type of cells in Fig 3C and Fig 3D. There are also

two graphs which are not mentioned and should be labelled with E and F

Figure 6, Figure 7, Figure 8 & Figure 9: What does the authors mean by “vehicle”?

6. PLOS authors have the option to publish the peer review history of their article (what does this mean?). If published, this will include your full peer review and any attached files.

Reviewer #1: No

Reviewer #2: No

Reviewer #3: **Yes: **Mazen Alzaharna

---

## [Author Response · Author response to Decision Letter 0]

5 Jun 2022

Response to editor and reviewer comments 

Response: done. 

Thank you for stating the following in the Acknowledgments Section of your manuscript: 

This work was supported by Kuwait University Research Sector grant PT02/18. Parts of this work were supported by grant SRUL02/13 to the Research Unit for Genomics, Proteomics and Cellomics Studies (OMICS), Kuwait University. 

This work was supported by Kuwait University Research Sector grant PT02/18. Parts of this work were supported by grant SRUL02/13 to the Research Unit for Genomics, Proteomics and Cellomics Studies (OMICS), Kuwait University. 

the funders had no role in study design, data collection and analysis, decision to publish, or preparation of the manuscript.

Response: The funding statement is now removed from the acknowledgment section in the revised version of the manuscript. Please keep the funding statement as it is in the online submission form as we are required by our Funder to acknowledge their support. 

Please amend your list of authors on the manuscript to ensure that each author is linked to an affiliation. Authors’ affiliations should reflect the institution where the work was done (if authors moved subsequently, you can also list the new affiliation stating “current affiliation:….” as necessary).

Response: all the authors have the same affiliation which is: Faculty of Pharmacy, Kuwait University, Safat 13110, Kuwait. 

Review Comments to the Author

 Reviewer #1: 

The authors examined the effect of glucose deprivation on growth and proliferation of breast cells, and the potentiation of Glucose deprivation on anti-proliferative effects of paclitaxel and doxorubicin. The novelty of this manuscript should be improved by exploring the underlying mechanism. As such, it still has some clinical value for drug treatment of tumors. A few of suggestions are shown below:

1. If possible, the author should provide some lines of supportive evidence from animal experiments. However, it is hard to set up a low-glucose condition because tumor cells have a strong ability of glucose uptake to enter the glycolysis.

Response: We agree with the reviewer regarding the clinical value of our presented data in terms of reducing the dose of chemotherapeutic agents and their dose-dependent side effects when combined with low-glucose regimen. Our results suggest that it may not be necessary to induce any severe form of glucose deprivation in vivo but just a substantial reduction, which can be achieved. Regarding the underlying mechanism, we have actually discussed this aspect in the last paragraph of the Discussion section of the manuscript citing studies using ketogenic diet. Please see below: 

As a therapeutic approach, depriving the body completely of glucose would of course have serious physiological consequences but lowering it temporarily only whilst chemotherapy was being administered might be tolerated. One way to achieve this situation is through the use of ketogenic diet (KD) for a short period of time. The effect of KD on metabolic parameters is variable depending on the composition of protein, fat, and carbohydrates as well as on the duration of its use. Some reports suggested that KD was shown to reduce fasting blood glucose and HbA1c levels in patients with diabetes [1-4], in normal subjects [5] and in rodents [6]. Interestingly, KD was also shown to reduce glucose metabolism in rodents and humans [7-9]. There is extensive clinical evidence indicating that KD is well tolerated in patients with various forms of cancer, resulting in improvement in quality of life with no serious adverse events or toxicity reported [10-23]. KD was shown to enhance the anti-tumor effects of various agents as well as when combined with radiotherapy. Combination of KD with bevacizumab in an orthotopic U87MG glioblastoma model in nude mice increased survival rate when compared with bevacizumab monotherapy[24]. Two reports suggested that KD significantly enhanced the anti-tumor and anti-angiogenic effects of metronomic cyclophosphamide in neuroblastoma xenografts in a CD1-nu mouse model [25, 26]. Furthermore, another report demonstrated that KD combination with radiation or carboplatin led to slower tumor growth in mice bearing NCI-H292 and A549 lung cancer xenografts; in part through increased oxidative damage mediated by lipid peroxidation [27]. KD was also shown to improve responses to several PI3K inhibitors in tumors with a wide range of genetic aberrations such as in patient-derived xenograft models of advanced endometrial adenocarcinoma (harboring a PTEN deletion and PIK3CA mutation), bladder cancer (FGFR-amplified), in syngeneic allograft models of PIK3CA mutant breast cancer and in a MLL-AF9 driven acute myeloid leukemia. Combination of KD improved the efficacy of several agents which target the PI3K pathway through inhibition of insulin feedback (which limits the efficacy of these agents) in part through decreased cell proliferation and increased apoptosis [28].

We also showed that glucose deprivation did not enhance cell apoptosis as indicated in Supplementary Figure 1, suggesting that reduced cell growth under deprived glucose conditions is not due to enhanced apoptotic machinery. In addition, we have published a paper in Frontiers of Pharmacology providing preliminary evidence for the signaling molecules which are modulated upon lactate supplementation to breast cancer cells; this is also related to glucose metabolism (PMID: 34744727). 

We are currently initiating further studies in which we aim to perform genomic and proteomic profiling of various markers involved in breast cancer cell proliferation, motility, and invasion to be tested under conditions of glucose or lactate deprivation, and upon exogenous lactate supplementation to breast cancer cells to better understand the underlying mechanism related to this phenomenon. 

2. In Figure 8, different concentrations of 2-DG treatment caused the similar effects to those obtained from Glucose deprivation. Also, addition of lactate or pyruvate could offset the effect of glucose deprivation. The authors should provide some mechanistic evidence to support their conclusion.

Response: Presumably the reviewer is referring to Figures 6 and 7 which show 2-DG treatment dose-dependently reduced breast cancer cell proliferation and motility. As indicated in the results section of the manuscript,

Effect of 2-deoxy glucose on cell proliferation and cell motility

2-DG is a D-glucose analogue which inhibits glycolysis through formation and intracellular accumulation of 2-deoxy-D-glucose-6-phosphate, a competitive inhibitor of hexokinase and glucose-6-phosphate isomerase [29]. 

We used 2-DG to confirm by another method [other than taking glucose out of the culture medium (as shown in Figures 1 and 5)] that glucose deprivation reduced breast cancer cell proliferation and motility. This was a confirmatory experiment to support our findings. We provided mechanistic explanation in the Discussion section of the manuscript of how glucose deprivation can reduce cell motility and proliferation. Please see below: 

Several previous reports have demonstrated similar effects of 2-DG on breast cancer cells, with reduction in their migration [30] and proliferation [31-35]. In another recent report, enhanced glucose uptake was correlated with enhanced invasiveness of several breast cancer cell lines, whereas low doses of 2-DG (1 mM) significantly reduced their invasive capacity [36]. Our data is consistent with these reports, in the same concentration range (0.5-1 mM). Chen et al [37] reported that short term glucose deprivation (24 h) in various breast cancer cell lines induced cell death, which was higher in MDA-MB-231 than MCF-7 cells, in part through enhanced AMPK phosphorylation. This was also observed in another report using MCF-7 and T47D cells [38]. We did not see much difference in cell proliferation at 24 h; it was evident at day 4 of glucose starvation in both the cancer lines as well as the normal cells. Periodic re-supply of glucose (every 72 h; Fig 1 E-F) maintained the cells at the seeding number for MCF10A and facilitated partial growth for pII cells. 

We also showed that glucose deprivation did not enhance cell apoptosis as indicated in Supplementary Figure 1, suggesting that reduced cell growth under deprived glucose conditions is not due to enhanced apoptotic machinery. In addition, we have a recent publication in Frontiers of Pharmacology providing preliminary evidence for the signaling molecules which are modulated upon lactate supplementation to breast cancer cells; this is also related to glucose metabolism (PMID: 34744727). We showed that cells treated with lactate (20 mM) had increased phosphorylation of ERK1/2 but did not show any change in either p38 MAPK or AKT phosphorylation. Also, the expression profile of focal adhesion kinase (FAK) was not modulated by lactate treatment. Interestingly, E-cadherin expression was significantly reduced by lactate treatment which might lead to loss of cell-cell connection and enhanced degree of motility, which was observed in cancer cells upon lactate supplementation (PMID: 34744727). 

We are currently initiating further studies in which we aim to perform genomic and proteomic profiling of various markers involved in breast cancer cell proliferation, motility, and invasion to be tested under conditions of glucose or lactate deprivation, and upon exogenous lactate supplementation to breast cancer cells to better understand the underlying mechanism related to this phenomenon. 

Reviewer #2: 

The manuscript: Glucose deprivation reduces proliferation and motility, and enhances the anti-proliferative effects of paclitaxel and doxorubicin in breast cell line in vitro, brings information already available in the literature on the effects of glucose metabolism on tumor and normal cell proliferation. Tumor cells produce a high level of reactive oxygen species compared to normal cells due to the increased activation of various metabolic pathways. Glycolysis by the Warburg effect maintains redox homeostasis by being independent of mitochondrial oxidative phosphorylation that produces a large amount of ROS, increasing metabolic secondary pathways, NADPH, G6PD and 6-Phosphogluconate dehydrogenase, which produces an effect on tumor metabolic reprogramming.

The data presented on the cytotoxic activity of the conditions of glucose deprivation, addition of lactate, DG2gly, should be normalized, since the data are presented in optical density values and do not represent the biological effects after the different experimental conditions. Thus, information about the effects of cytotoxicity can be better interpreted and included in the discussion.

Response: Presumably the reviewer is highlighting the manner in which results for MTT assay are presented in Figures 1, 2, and 6. It is quite common practice for MTT data to be presented directly as OD values rather than converting into cell numbers which would need a standard curve each time with different cell numbers. This way of presentation would only change the units in the Y axis (from OD to cell number) but the histobars would be exactly the same relative to each other. We have published several manuscripts where we described MTT data in terms of OD readings (PMID: 34744727, PMID: 31980706, PMID: 30365135, PMID: 28276317, PMID: 26718772). Examples of recent papers where other authors have similarly presented MTT data as OD readings are PMID: 34951409, PMID: 34929154, PMID: 34916844.

Having said that, we have actually also included cell number information at Day 0 (hatched bar) in Figure 1 A-D, and Figure 6 A-D, and at Day 0 in Figure 1 E-F where the number of cells were measured using a hemocytometer as described in the Methods section. 

Regarding MTT data presented in Figures 8-9, we measured the degree of inhibition of cell proliferation by paclitaxel or doxorubicin when cells were cultured in medium with no added glucose, low glucose 1.7-5 mM, or high glucose (+ glucose medium, 25 mM). Since the degree of cell proliferation is variable when the cells were cultured in medium with different glucose concentrations (1.7, 5, or 25 mM), we therefore normalized these data by taking the vehicle as 100% for each condition and then compared the degree of inhibition of cell proliferation induced by the addition of either paclitaxel or doxorubicin with its own vehicle. These results are extensively discussed in the manuscript. Please see below: 

Glucose depletion significantly enhanced the anti-proliferative effects of paclitaxel and doxorubicin in breast cancer cell lines

We wanted to determine if glucose depletion enhances the sensitivity of paclitaxel or doxorubicin in inhibiting cell proliferation. Paclitaxel (Fig 8) and doxorubicin (Fig 9) treatment significantly reduced cell proliferation in a concentration dependent manner. There was a significant inhibition in pII cell proliferation at 1 µM paclitaxel, which was further increased to 90% with 10-100 µM (Fig 8 A). For MDA-MB-231 cells, paclitaxel inhibited cell proliferation (by 50%) with 0.1 µM, which reached 98-99% inhibition with 1-100 µM (Fig 8 B). For YS 1.2 cells, paclitaxel (0.1-100 µM) inhibited cell proliferation by 80 % (Fig 8 C). For MCF10A, 20-25% inhibition was seen with paclitaxel at concentrations of 0.1-1 µM and reached 50% inhibition at concentrations of 10-100 µM (Fig 8 D). Of note, breast cancer cells were more sensitive to paclitaxel treatment when compared to the normal epithelial cell line MCF10A. Doxorubicin significantly inhibited pII cell proliferation (by 30%) at a concentration of 100 µM (Fig 9 A). For MDA-MB-231 cells, doxorubicin significantly inhibited cell proliferation (by 80-99%) at all the concentrations used (Fig 9 B). These data suggest that de novo resistant breast cancer cells are more sensitive to the anti-proliferative effect of doxorubicin when compared to the acquired endocrine resistant cells. For YS 1.2, there was a significant inhibition in cell proliferation (by 50%) at 1 µM, which further increased to 70-80% with 10-100 µM (Fig 9 C). For MCF10A, there was a significant inhibition of cell proliferation (by 40-50%) at the concentrations used (Fig 9 D). 

To determine whether a synergistic/additive effect could be obtained in low glucose medium, we added these drugs to cells cultured in either medium without glucose or supplemented with low (1.7 or 5 mM) glucose. When the tested cell lines were cultured in medium without glucose (for 4 days), there was no evidence of cell proliferation; adding either paclitaxel or doxorubicin was of no benefit (data not shown). Therefore, we next determined the effect of these two agents in a medium supplemented with low glucose. 

For pII cells, glucose reduction significantly enhanced the anti-proliferative effects of both paclitaxel and doxorubicin when compared to treatment in the presence of glucose. The addition of 1.7 mM glucose had a greater enhancement effect than the higher amount. In the reduced glucose medium, treatment with 0.1 µM paclitaxel induced a broadly comparable degree of inhibition to that achieved with 100 µM in the presence of glucose in the culture medium (Fig 8 A). Also, in reduced glucose medium, treatment with 0.1 µM of doxorubicin induced a greater degree of inhibition than 100 µM concentration in the presence of glucose in the culture medium (Fig 9 A). 

For MDA-MB-231 cells, glucose reduction enhanced the anti-proliferative effect of paclitaxel (but not doxorubicin) at 0.1 µM (Fig 8 B and 9 B) and higher concentrations of both agents abolished cell proliferation in all of the tested conditions. For YS1.2, higher degree of inhibition at all paclitaxel concentrations was achieved in cells cultured in glucose reduced medium (Fig 8 C), and a similar degree of inhibition with doxorubicin was achieved with 0.1 µM in glucose reduced medium compared with 100 µM in glucose containing medium (Fig 9 C). Similar effects were also seen with MCF10A cells (Fig 8 D and 9 D). 

The data presented in the supplementary material does not demonstrate the antiproliferative effects of the metabolic effects of glucose deprivation, since no increase in cells in apoptosis or necrosis was found.

Response: 

Inhibition of proliferation is simply blocking further growth. It does not have to involve cell death (i.e. apoptosis). We have provided mechanistic explanations in the Discussion section of the manuscript for how glucose deprivation can reduce cell motility and proliferation. Please see below: 

Several previous reports have demonstrated similar effects of 2-DG on breast cancer cells, with reduction in their migration [30] and proliferation [31-35]. In another recent report, enhanced glucose uptake was correlated with enhanced invasiveness of several breast cancer cell lines, whereas low doses of 2-DG (1 mM) significantly reduced their invasive capacity [36]. Our data is consistent with these reports, in the same concentration range (0.5-1 mM). Chen et al [37] reported that short term glucose deprivation (24 h) in various breast cancer cell lines induced cell death, which was higher in MDA-MB-231 than MCF-7 cells, in part through enhanced AMPK phosphorylation. This was also observed in another report using MCF-7 and T47D cells [38]. We did not see much difference in cell proliferation at 24 h; it was evident at day 4 of glucose starvation in both the cancer lines as well as the normal cells. Periodic re-supply of glucose (every 72 h; Fig 1 E-F) maintained the cells at the seeding number for MCF10A and facilitated partial growth for pII cells. 

We also showed that glucose deprivation did not enhance cell apoptosis as indicated in Supplementary Figure 1, suggesting that reduced cell growth under deprived glucose conditions is not due to enhanced apoptotic activity. In addition, we have a recent publication in Frontiers of Pharmacology providing preliminary evidence for the signaling molecules which are modulated upon lactate supplementation to breast cancer cells; this is also related to glucose metabolism (PMID: 34744727). 

We are currently initiating further studies in which we aim to perform genomic and proteomic profiling of various markers involved in breast cancer cell proliferation, motility, and invasion to be tested under conditions of glucose or lactate deprivation, and upon exogenous lactate supplementation to breast cancer cells to better understand the underlying mechanism related to this phenomenon. 

Include in the discussion other aspects of non-glycolytic metabolism (Non-glycolysis Glucose Metabolism Pathway in Breast Cancer). 

Response: The following paragraph is now included in the Discussion section of the revised version of the manuscript to mention these. Please see below:

As already discussed, cancer cells mainly utilize aerobic glycolysis as the main route of glucose metabolism after glucose entry into cells through glucose transporters. This results in the generation of various glucose metabolites which may enter other metabolic pathways such as the serine/glycine and pentose phosphate metabolic pathways (termed non-glycolysis metabolic pathways). All the products generated through these various pathways play an important role in cancer metabolism, progression, and metastasis [39, 40]. The pentose shunt provides precursors for nucleic acid biosynthesis that is important for cancer cells to proliferate. It was demonstrated that the expression profile of serine/glycine metabolic pathway is unique to the molecular subtype of breast cancer, with high activity in HER-2 [41] and triple negative types [42], which is correlated with poor clinical prognosis [43]. In addition, enhanced expression profile of the pentose phosphate pathway-related enzymes such is 6PGDH and TKT is also evident in breast cancer [44, 45], especially in HER-2 and triple negative types [46], which is again associated with poor clinical prognosis [47-49].

The authors could apply a predictive pharmacological test to demonstrate the synergistic effects of chemotherapeutic combinations in deprivation or supplementation conditions, such as additive, antagonism or synergistic effects.

Response: It is not quite clear what the Reviewer means by “predictive pharmacological test” However, the MTT data presented in Figures 8-9 show the degree of inhibition in cell proliferation by paclitaxel or doxorubicin when cells were cultured in medium with no added glucose, low glucose 1.7-5 mM, or high glucose (+ glucose medium, 25 mM). Since the degree of cell proliferation was variable when the cells were cultured in medium with different glucose concentrations (1.7, 5, or 25 mM), we therefore normalized these data by taking the vehicle as 100% for each condition and then compared the degree of inhibition of cell proliferation induced by the addition of either paclitaxel or doxorubicin with its own vehicle. Since we normalized these data, we cannot calculate the synergistic/additive effects. We tried to represent these data in several different ways, but it became too complicated to present, and this way was the most understandable and clear to the reader. 

Reviewer #3: 

Minor Issues:

Materials and methods- Cell lines- second paragraph:

The authors wrote 100 U/Ml penicillin should be corrected to 100 U/mL

Response: done. 

RPMI medium used (Cat# R1383) contains L-glutamine (0.3 g/L) and you also added L-glutamine as a supplement. This could be another source of energy for the cells. L-glutamine serves as an auxiliary energy source, especially when cells are rapidly dividing. Why the non-essential amino acids were added to the culture medium? Published research reported that non-essential amino acids attenuate apoptosis.

Response: It is not uncommon practice to add supplements to the culture medium for growth of cell lines to optimize their growth, and it is something we do routinely. Sometimes we only add L-glutamine and not the amino acids and it does not seem to make much difference. Regarding LO-glutamine, we have indeed looked to see whether it can support cell growth in low glucose conditions but our experiments in this regard have been rather inconclusive and we have no clear indication that L-glutamine can serve as an alternative energy source although that was our initial expectation. So, we did not alter our culture conditions in that regard as cells do depend on L-glutamine for other nutritional purposes.

According to the manufacturer’s instructions the concentration of cells for Annexin V apoptosis experiment is 1 x 10*6 cells/ml, why the authors used a concentration of 5 x 10*6 cells/ml.

Response: This was a typographical error. This is now corrected in the new version of the manuscript (1x106 cells/ml).

Why the authors did not use the same medium for the experiments with glucose and those without glucose?

Response: We routinely use DMEM, but we were unable to procure DMEM without glucose, and therefore we used RPMI medium that was available without glucose for the glucose deprivation experimental conditions. In fact, we have compared cell behavior when cultured in either DMEM or RPMI media and there appears to be no significant difference. 

Figure 1: To denote a significant difference, please insert a line above the bars that are targeted and add a * above the line. This will make it easier for the readers to follow. I think the X axis in the figure can be labelled in a more appropriate way.

Response: A line above the bars is now included in the NEW Fig 1. We would like to keep the X axis label as it is for consistency with the other figures in the manuscript. 

Figure 3: It is not mentioned in the legend and neither on the figure the type of cells in Fig 3C and Fig 3D. There are also two graphs which are not mentioned and should be labelled with E and F.

Response: all the suggested changes are now included in the NEW Fig 3, and Fig 3 legend. 

Figure 6, Figure 7, Figure 8 & Figure 9: What does the authors mean by “vehicle”?

Response: “Vehicle” is the common term used to refer to the diluent used to dissolve/dilute drugs. In this case for 2-DG, doxorubicin, or paclitaxel the vehicle was saline. This is now clarified in Fig 7 and 8 legends in the revised version of the manuscript. 

Note: We also recognized a typo in the Y-axis in Fig 7, panel B. This is now corrected in the NEW Fig 7.

---

## [Decision Letter · Decision Letter 1]

21 Jun 2022

PONE-D-22-03147R1Glucose deprivation reduces proliferation and motility, and enhances the anti-proliferative effects of paclitaxel and doxorubicin in breast cell line in vitroPLOS ONE

Dear Dr. Khajah,

Thank you for submitting your manuscript to PLOS ONE. After careful consideration, we feel that it has merit but does not fully meet PLOS ONE’s publication criteria as it currently stands. Therefore, we invite you to submit a revised version of the manuscript that addresses the points raised during the review process.

We look forward to receiving your revised manuscript.

Kind regards,

Yi-Hsien Hsieh, Ph.D.

Academic Editor

PLOS ONE

Reviewers' comments:

Reviewer's Responses to Questions

**Comments to the Author**

1. If the authors have adequately addressed your comments raised in a previous round of review and you feel that this manuscript is now acceptable for publication, you may indicate that here to bypass the “Comments to the Author” section, enter your conflict of interest statement in the “Confidential to Editor” section, and submit your "Accept" recommendation.

Reviewer #1: All comments have been addressed

Reviewer #2: All comments have been addressed

Reviewer #3: All comments have been addressed

2. Is the manuscript technically sound, and do the data support the conclusions?

Reviewer #1: Yes

Reviewer #2: Partly

Reviewer #3: Yes

3. Has the statistical analysis been performed appropriately and rigorously? 

Reviewer #1: Yes

Reviewer #2: No

Reviewer #3: Yes

4. Have the authors made all data underlying the findings in their manuscript fully available?

Reviewer #1: Yes

Reviewer #2: Yes

Reviewer #3: Yes

5. Is the manuscript presented in an intelligible fashion and written in standard English?

Reviewer #1: Yes

Reviewer #2: No

Reviewer #3: Yes

6. Review Comments to the Author

Reviewer #1: The authors have adequately modified their manuscript. Besides, they also carefully answers the questions. Thereby, this paper could be acceptable for publication in this journal.

Reviewer #2: The authors partially performed the suggested corrections, but the presentation of the cytotoxicity graphs compromises the veracity of the data due to the different optical densities existing between the cells.

The initial difference between the control groups exceeds more than 40%, as suggested the data should be normalized.

The different optical densities found in the graphs reflect the metabolic differences between tumor cell types.

After normalization of optical densities, the values must be expressed in Percentage of Viability (%) in relation to the group of untreated (control cells).

I suggest using a program to normalize the data, for example the Graphic Pad Prism, or similar

Normalize the data to convert Y values from different data sets to a common scale. If you can't get Normalize to do what you want, take a look at the Remove Baseline analysis which can do some kinds of normalizing.

https://www.graphpad.com/scientific-software/prism/

Authors must correct the data presented in the way they are described, compromising their veracity

Reviewer #3: The authors have responded to the addressed comments afrom my sidend corrected the manuscript accordingly. No other comments

7. PLOS authors have the option to publish the peer review history of their article (what does this mean?). If published, this will include your full peer review and any attached files.

Reviewer #1: No

Reviewer #2: **Yes: **Prof. Durvanei Augusto Maria

Reviewer #3: No

---

## [Author Response · Author response to Decision Letter 1]

29 Jun 2022

Response to reviewer comments

Reviewer #1: The authors have adequately modified their manuscript. Besides, they also carefully answer the questions. Thereby, this paper could be acceptable for publication in this journal.

Response: we thank the reviewer for the positive comment. 

Reviewer #2: The authors partially performed the suggested corrections, but the presentation of the cytotoxicity graphs compromises the veracity of the data due to the different optical densities existing between the cells. The initial difference between the control groups exceeds more than 40%, as suggested the data should be normalized. The different optical densities found in the graphs reflect the metabolic differences between tumor cell types. After normalization of optical densities, the values must be expressed in Percentage of Viability (%) in relation to the group of untreated (control cells).

I suggest using a program to normalize the data, for example the Graphic Pad Prism, or similar

Normalize the data to convert Y values from different data sets to a common scale. If you can't get Normalize to do what you want, take a look at the Remove Baseline analysis which can do some kinds of normalizing.

https://www.graphpad.com/scientific-software/prism/

Authors must correct the data presented in the way they are described, compromising their veracity

Response: We thank the reviewer for the valid comments. As per the recommendation we have modified the data presented in Fig 8 and 9 by considering the variability in the seeding density for vehicle treated cells with different glucose concentration in the culture medium. In the NEW modified Figs 8 and 9, vehicle treated cells cultured in glucose containing medium (25 mM, Black lines) is taken as 100% and the data are normalized accordingly. 

The appropriate wording changes have been made in the ‘Results’ and ‘figure legends’ sections in the marked version of the manuscript. 

We trust this clarifies the data and is what the Reviewer had in mind to be done. 

Reviewer #3: The authors have responded to the addressed comments from my side and corrected the manuscript accordingly. No other comments

Response: we thank the reviewer for the positive comment.

---

## [Decision Letter · Decision Letter 2]

5 Jul 2022

PONE-D-22-03147R2Glucose deprivation reduces proliferation and motility, and enhances the anti-proliferative effects of paclitaxel and doxorubicin in breast cell line in vitroPLOS ONE

Dear Dr. Khajah,

Thank you for submitting your manuscript to PLOS ONE. After careful consideration, we feel that it has merit but does not fully meet PLOS ONE’s publication criteria as it currently stands. Therefore, we invite you to submit a revised version of the manuscript that addresses the points raised during the review process.

We look forward to receiving your revised manuscript.

Kind regards,

Yi-Hsien Hsieh, Ph.D.

Academic Editor

PLOS ONE

Journal Requirements:

Reviewers' comments:

Reviewer's Responses to Questions

**Comments to the Author**

1. If the authors have adequately addressed your comments raised in a previous round of review and you feel that this manuscript is now acceptable for publication, you may indicate that here to bypass the “Comments to the Author” section, enter your conflict of interest statement in the “Confidential to Editor” section, and submit your "Accept" recommendation.

Reviewer #2: All comments have been addressed

2. Is the manuscript technically sound, and do the data support the conclusions?

Reviewer #2: No

3. Has the statistical analysis been performed appropriately and rigorously? 

Reviewer #2: Yes

4. Have the authors made all data underlying the findings in their manuscript fully available?

Reviewer #2: Yes

5. Is the manuscript presented in an intelligible fashion and written in standard English?

Reviewer #2: Yes

6. Review Comments to the Author

Reviewer #2: To the authors of the manuscript, sorry for the insistence, the cell viability data must be corrected.

In the forwarded version, no changes were made to the figures.

Therefore, I do not recommend publication, since the data presented were not analyzed and presented properly.

7. PLOS authors have the option to publish the peer review history of their article (what does this mean?). If published, this will include your full peer review and any attached files.

Reviewer #2: No

---

## [Author Response · Author response to Decision Letter 2]

6 Jul 2022

Response to editor and reviewer comments 

Journal Requirements:

Response: We do not understand the editorial remark concerning the references, as no issue was raised by any of the Reviewers in this context. Is this perhaps a new routine requirement by the journal to ensure that no retracted papers are cited?. We have double-checked all the references cited in this manuscript and to the best of our knowledge they are correctly cited and do not include any retracted papers that we are aware of. We actually checked each cited paper in PubMed and none of them were retracted. 

Reviewer #2: 

Old comments:

The authors partially performed the suggested corrections, but the presentation of the cytotoxicity graphs compromises the veracity of the data due to the different optical densities existing between the cells. The initial difference between the control groups exceeds more than 40%, as suggested the data should be normalized. The different optical densities found in the graphs reflect the metabolic differences between tumor cell types. After normalization of optical densities, the values must be expressed in Percentage of Viability (%) in relation to the group of untreated (control cells).

I suggest using a program to normalize the data, for example the Graphic Pad Prism, or similar

Normalize the data to convert Y values from different data sets to a common scale. If you can't get Normalize to do what you want, take a look at the Remove Baseline analysis which can do some kinds of normalizing.

New comments:

To the authors of the manuscript, sorry for the insistence, the cell viability data must be corrected.

In the forwarded version, no changes were made to the figures. Therefore, I do not recommend publication, since the data presented were not analyzed and presented properly.

Response: 

We are very surprised by the reviewer’s comments regarding the cell viability data. In fact, we have carefully addressed all the comments previously raised by this and the other reviewers. We have responded to Reviewer two's comments regarding cell viability data presentation in Figures 8 and 9. There is some misunderstanding as we clearly submitted corrected figures as required by this reviewer. We are including in this letter the old and new versions of both figures to demonstrate the differences in the data presentation. We have considered the initial difference between the control (vehicle) groups cultured with different glucose concentration in the culture medium. In the NEW modified Figs 8 and 9, vehicle treated control cells cultured in glucose containing medium (25 mM, Black lines) is taken as 100% and the data are normalized accordingly to a common scale as the Reviewer suggested. This was our understanding of what the Reviewer had requested, and we have complied with it. 

Although we do not perceive this to be the case, but if the continued objection is only the fact that the y axis is labelled as OD instead of cell number (?), we would point out that it is quite common practice for MTT data to be presented directly as OD values. We have published several manuscripts where we described MTT data in terms of OD readings (PMID: 34744727, PMID: 31980706, PMID: 30365135, PMID: 28276317, PMID: 26718772). Examples of recent papers where other authors have similarly presented MTT data as OD readings are PMID: 34951409, PMID: 34929154, PMID: 34916844. 

But if there is something else that we have missed in his comment we would be grateful if the Reviewer could explain it more precisely and we would be happy to address the concern.

---

## [Editor Report · Decision Letter 3]

20 Jul 2022

Glucose deprivation reduces proliferation and motility, and enhances the anti-proliferative effects of paclitaxel and doxorubicin in breast cell line in vitro

PONE-D-22-03147R3

Dear Dr. Khajah,

We’re pleased to inform you that your manuscript has been judged scientifically suitable for publication and will be formally accepted for publication once it meets all outstanding technical requirements.

Kind regards,

Yi-Hsien Hsieh, Ph.D.

Academic Editor

PLOS ONE

Additional Editor Comments (optional):

All comments have been addressed.
---

## [Editor Report · Acceptance letter]

22 Jul 2022

PONE-D-22-03147R3 

Glucose deprivation reduces proliferation and motility, and enhances the anti-proliferative effects of paclitaxel and doxorubicin in breast cell line in vitro 

Dear Dr. Khajah:

I'm pleased to inform you that your manuscript has been deemed suitable for publication in PLOS ONE. Congratulations! Your manuscript is now with our production department. 

Kind regards, 

on behalf of

Dr Yi-Hsien Hsieh 

Academic Editor

PLOS ONE